# Male-Type and Prototypal Depression Trajectories for Men Experiencing Mental Health Problems

**DOI:** 10.3390/ijerph17197322

**Published:** 2020-10-07

**Authors:** Simon M. Rice, David Kealy, Zac E. Seidler, John L. Oliffe, Ronald F. Levant, John S. Ogrodniczuk

**Affiliations:** 1Orygen, Parkville, Melbourne 3052, Australia; zac.seidler@orygen.org.au; 2Centre for Youth Mental Health, The University of Melbourne, Melbourne 3052, Australia; 3Department of Psychiatry, University of British Columbia, Vancouver, BC V6T 1Z4, Canada; david.kealy@ubc.ca (D.K.); john.ogrodniczuk@ubc.ca (J.S.O.); 4School of Nursing, University of British Columbia, Vancouver, BC V6T 2B5 Canada; john.oliffe@ubc.ca; 5Department of Nursing, The University of Melbourne, Melbourne, Victoria 3010, Australia; 6Department of Psychology, University of Akron, Akron, OH 44325-4301, USA; levant@uakron.edu

**Keywords:** depression, assessment, gender, men, latent growth curve

## Abstract

Growing interest in gender-sensitive assessment of depression in men has seen the development of male-specific screening tools. These measures are yet to be subject to longitudinal latent modelling, which limits evidence about the ability of these tools to detect change, especially relative to established screening scales. In this study, three waves of data were collected from 234 men (38.35 years, SD = 14.09) including 3- and 6-month follow-up. Analyses focused on baseline differences and symptom trajectories for the Patient Health Questionnaire (PHQ; prototypic symptoms) and the Male Depression Risk Scale (MDRS; male-type symptoms). At baseline, men not accessing treatment reported higher MDRS scores relative to treatment-engaged men. There was no group difference for the PHQ. Internal consistency (α, ω) coefficients indicated comparable reliability for both measures across the three waves. Multidomain latent growth modelling, including current treatment engagement as a covariate, reported good model fit (CFI = 0.964, TLI = 0.986, RMSEA = 0.081, SRMR = 0.033) with differential findings for the PHQ and MDRS. Consistent with the baseline between-group analysis, current treatment effects were observed for the MDRS, but not the PHQ. Trajectory modelling for the MDRS indicated that greater severity resulted in slower improvement by 6 months. In contrast, there was no difference in the PHQ rate of change between baseline and 6 months. Findings support the psychometric utility of the MDRS as a male-specific symptom domain measure sensitive to both longitudinal change and potential treatment effects for symptomatic men, in ways not discernible by the PHQ. The MDRS may be a useful adjunctive screening tool for assessing men’s depression.

## 1. Introduction

Improved assessment of male depression is gaining momentum internationally as a means of reducing male suicide, and the construct of a distinct clinical phenotype is central to this work [1,2,3]. Meta-analytic research shows that depression is a significant risk factor for suicide [4], with both male gender and the misuse of alcohol or drugs as important predictors [5]. Underscoring the gendered nature of the problem, worldwide, suicide occurs 1.8-times more frequently among men than women [6]. The growing recognition of suicide as a gendered phenomenon has led to a greater focus on risk factors experienced by men [7,8]. Building on early qualitative work introducing the possibility of a unique profile of externalising and male-type symptoms experienced by some depressed men [9,10,11,12,13,14], practitioners have developed and validated a number of male-specific depression screening tools [15,16,17,18,19]. Currently available male-specific measures have sought to assess a broader range of symptoms (relative to prototypic depression measures) that align with men’s socialisation and gender norm processes. For example, emotional restrictiveness and self-reliance are often promoted among men [20], while externalising behaviours (anger, aggression, alcohol use) may be condoned as responses to male distress [21,22,23]. In seeking to assess these broader domains, male-specific depression measures include symptoms assessing anger and irritability, substance misuse, risk-taking and recklessness, and non-externalising manifestations, including emotion suppression and somatic symptoms—all of which largely fall outside the prototypic symptoms of major depressive disorder [24]. Whereas use of male-specific depression measures continues to grow, psychometric studies are lacking [25].

To date, the most widely validated of the currently available male-specific tools is the Male Depression Risk Scale (MDRS; [26]). Developed using exploratory and confirmatory factor analysis, the MDRS has shown test-retest stability [27], in addition to good sensitivity in detecting men’s suicide risk and cross-nation factor structure stability [28]. The MDRS progresses the early pioneering work of the Gotland research programme into men’s depression, established in Sweden in the 1990s. Whereas the male-type depression syndrome, as originally articulated by Rutz et al. [17], has become a topic of significant interest in the men’s mental health literature [2], conclusive studies are yet to categorically support or refute the construct, with debate enduring [29].

One of the main methodological challenges for advancing the field is the predominance of cross-sectional studies and absence of men’s depression symptom trajectories modelled over time [30]. In seeking to address this gap, foundational work by Rice at al. [27] examined male depression trajectories over 16 weeks relative to stressful life events. In comparison to females, males experiencing stressful life events reported elevated MDRS scores. However, while these early results provided some supporting evidence for a differential male depression symptom trajectory, the design was limited by data collection at only two time points, precluding opportunities for complex modelling to account for within and between person changes beyond simple group means. 

Using three waves of data (baseline, 3 months, 6 months), we examined a multiple-group (e.g., current treatment yes/no), multiple-domain latent growth model (MDLGM) comparing longitudinal trajectories for the MDRS and Patient Health Questionnaire–Depression Module (PHQ; [31]), a widely used screening tool of the nine criterion symptoms contributing to a diagnosis of major depression [24]. The present study had four overarching aims. First, the study aimed to provide psychometric reliability data on the MDRS, benchmarked against the PHQ, to determine the relative longitudinal internal consistency values using omega and alpha coefficients across the three waves (Aim 1). Second, the study aimed to evaluate structural equation model fit indices for a multiple-group, multiple-domain latent growth model approach including treatment as a covariate predictor (Aim 2). Third, the study sought to determine whether baseline differences existed on PHQ and MDRS scores according to whether men self-reporting a mental health problem were or were not accessing treatment at baseline (Aim 3). Finally, the study aimed to assess if trajectories of change for PHQ and MDRS varied as a function of experiencing mental health problems either with or without treatment engagement (Aim 4). 

## 2. Materials and Methods

### 2.1. Participants and Design

Using a longitudinal design, three waves of data were captured, collected online at baseline, 3 months and 6 months between October 2018 and March 2019. Data for the present analysis were obtained from a larger sample of 3769 men who provided baseline data, 74.8% (*n* = 2819) of which self-reported having a mental health problem. Of the respondents reporting a mental health problem, a subset of 8.3% (*n* = 234) provided data at both wave 2 (3 months) and wave 3 (6 months), and were the focus of analysis for the present study. 

### 2.2. Measures

Male Depression Risk Scale (MDRS; [26]): The MDRS is a 22-item self-report scale designed to assess externalising and male-specific depression symptoms. Validation studies have supported the six-domain factor structure of the MDRS, assessing emotion suppression, drug use, alcohol use, anger and aggression, somatic symptoms and risk-taking [26,28]. Respondents rate items relative to the preceding month. The response options provide a descriptor and reference frequency range. These were: Not at all (0 days), occasionally (1–4 days), around 25% of the time (5–8 days), just under half the time (9–12 days), just over half the time (13–16 days), around 75% of the time (17–20 days), very frequently (21–24 days) and almost always (25+ days). Each item is scored from 0–7, respectively, with higher scores reflecting more frequent symptom experience. The scale and response format is presented in Appendix A. Test-retest reliability and convergent validity of the MDRS have been supported [27].

Patient Health Questionnaire–Depression Module (PHQ; [31]): The PHQ consists of nine items that correspond to DSM-5 [24] symptoms of major depressive disorder, referring to the frequency of symptoms over the past two weeks. Scoring uses a four-point scale anchored by not at all (0) and nearly every day (3). A higher total score (the sum of all items) indicates greater depressive severity. The psychometric properties and assessment accuracy of the PHQ have been well-validated and tested against structured psychiatric interviews in multiple studies and meta-analyses [32], and it is widely used in clinical and research settings for both the continuous measurement of depressive symptom severity and categorical screening for major depressive disorder.

### 2.3. Procedure

Participants were recruited online via the *HeadsUpGuys* website (https://headsupguys.org). The *HeadsUpGuys* website was developed in Canada and has become a leading global resource providing tips, tools, information about professional services, and recovery stories to help men fight depression and prevent suicide. *HeadsUpGuys* launched in June 2015 and offers those visiting the site the opportunity to complete a self-check for depression symptoms, with a range of international sites linking to the *HeadsUpGuys* resource. More information on *HeadsUpGuys* is available elsewhere [33]. In the present study, men who expressed an interest in participating were taken to an independent survey site, which was hosted by Qualtrics, where they were presented with the informed consent page. The only inclusion criterion applied was that participants self-identified as male, with participation open to men residing anywhere in the world. Those providing informed consent to participate then completed the survey online. Participants who were willing to complete the survey again at 3- and 6-month follow-up were sent notifications at these respective times. Ethics approval for the study was granted by the Behavioural Research Ethics Board at the University of British Columbia (H17-01334).

### 2.4. Data Analysis

Descriptive statistics were used to characterise the sample. Data screening examined univariate and multivariate normality. Inferential tests (ANCOVA, MANCOVA) explored group differences according to current mental health treatment, controlling for age, which is known to influence MDRS scores [34], with partial eta-squared reported as estimates of effect size. Reliability coefficients were calculated using Cronbach’s alpha and McDonald’s omega with the SPSS *omega* macro [35], implementing closed-form approximation of loadings for McDonald’s omega [36].

Whereas traditional change-based analysis approaches (e.g., within-group ANOVA, regression-based approaches) examine group mean changes, they treat differences among individual participants as error variance, some of which likely contain valuable information about change. Instead, we used the MDLGM approach, which allows exploration of whether there are relations among the patterns in growth processes across domains. Such MDLGMs yield more power than multiple univariate growth models because the MDLGM approach incorporates additional information using the correlations among latent growth factors across domains [37]. By enabling tracking of individual differences in the slope and intercept trajectories, MDLGMs can also identify variables (e.g., covariates) that exert important effects on observed symptom change [38]. MDLGM approaches require a minimum of three waves of data. In the present study, growth curve models (with the PHQ and MDRS modelled simultaneously) were undertaken to determine whether change trajectories were associated, estimating the extent of covariation in the development of pairs of symptom domains [38]. Structural equation modelling was undertaken for assessing multivariate growth curve models following procedures outlined by Byrne [39].

We used the following criteria for acceptable model fit with the comparative fit index (CFI) and the Tucker Lewis index (TLI) >0.95, standardized root mean square residual (SRMR) <0.08, and root mean square error of approximation (RMSEA); <0.1) values, respectively [40]. Whereas it is argued that RMSEA values should be <0.08, e.g., [39], the RMSEA index is affected by sample size [41], and the present sample (*n* = 234) is considered on the low end for the application of SEM to latent growth curve analysis [42]. We tested for linearity by fitting an additional quadratic model and inspection of the corresponding chi-square value. Within-domain (intra-individual change) and between-domain (inter-individual change) covariance patterns were reported to determine latent growth trajectories. Models were estimated using maximum likelihood estimation. At the final stage of analysis, the baseline variable current treatment (yes/no) was modelled as a time-invariant covariate in order to determine if it was associated with differential baseline and change scores, and a subsequent model evaluated treatment at 3 and 6 months as time variant covariates. Interpretation was guided by squared multiple correlation (*R*^2^) coefficients, summarising the proportion of variance accounted for by the predictors [43]. Analyses were undertaken in SPSS 26.0 and AMOS 25.0, IBM Corp, New York, NY, USA.

## 3. Results

### 3.1. Sample Characters

On average, the participants were aged 38.35 years, standard deviation (SD) = 14.09 (range = 18–73). Men who were currently accessing treatment were, on average, 4.5 years older (*M* = 40.37, *SD* = 13.67) than those not currently in treatment (*M* = 35.69, *SD* = 14.27), *t*(232) = −2.54, *p* = 0.012), and tended to be in higher income brackets (χ^2^(5) = 11.85, *p* = 0.037). There were no group differences for sexual orientation (76.1% heterosexual; 10.7% homosexual; 11.1% bisexual), student status (22.2%), ethnicity (2.6% Aboriginal; 0.9%; African; 2.1% Asian; 0.9% Hispanic; 84.2% Caucasian; 6.0% multiple ethnicities; 3.4% other), relationship status (45.7% single; 29.1% married; 15.4% committed relationship; 6.0% divorced, 3.8% separated) or self-rated general health (5.1% excellent; 18.4% very good; 40.2% good; 30.3 fair; 6.0% poor). Among the 234 men self-reporting a mental health problem at baseline, 133 (56.8%) reported that they were currently accessing mental health treatment. This decreased to 78 (33.3%) and 71 (30.3%) at 3 and 6 months. Most participants resided in Canada (*n* = 138; 59.0%), with the remaining participants residing in the US (*n* = 42; 17.9%), UK (*n* = 18; 7.7%), Australia (*n* = 19; 8.1%) or elsewhere (*n* = 17; 7.3%).

### 3.2. Baseline Differences—PHQ, MDRS

Skewness and kurtosis values were all within the normal range ±2.0, supporting univariate normality, with multivariate normality established via elliptical plots [39]. Descriptive statistics for the individual MDRS items and MDRS and PHQ total scores are presented in Table 1.

Consistent with a help-seeking population, on average, at baseline, participants were in the ‘moderate depression’ range on the PHQ (*M* = 16.52, *SD* = 6.34) and the ‘elevated risk’ range for the MDRS (*M* = 45.00, *SD* = 21.10). Three MANCOVAs were conducted with baseline PHQ and MDRS items, and the six MDRS subscales as dependent variables, current treatment engagement as the independent variable and age as covariate. There was no multivariate effect observed for current treatment for the PHQ items (Λ = 0.947, *F*(9, 223) = 1.40, *p* = 0.191, partial η^2^ = 0.053). In contrast, there was a large multivariate effect for the 22 MDRS items (Λ = 0.835, *F*(22, 210) = 1.89, *p* = 0.012, partial η^2^ = 0.165), which attenuated to a moderate multivariate effect when the six MDRS subscales were evaluated as dependent variables (Λ = 0.931, *F*(6, 226) = 2.78, *p* = 0.012, partial η^2^ = 0.069). Age was not a significant covariate, at either the multivariate or univariate level, for any analysis. As can be seen from Table 1, three MDRS item scores were significantly higher for men not in treatment than those currently in treatment. At the MDRS subscale level, those not in treatment reported higher scores than those currently in treatment for the emotion suppression *F*(1, 231) = 8.91, *p* < 0.001, partial η^2^ = 0.037 and risk-taking domains *F*(1, 231) = 7.69, *p* = 0.006, partial η^2^ = 0.032. Finally, ANCOVAs were undertaken for the MDRS-22 and PHQ total scores. As shown in Table 1, higher baseline MDRS-22 scores (but not PHQ-9 scores) were observed for those not engaged in current treatment *F*(1, 231) = 6.35, *p* = 0.012, partial η^2^ = 0.027.

### 3.3. Internal Consistency and Correlations Across Waves

To examine Aim 1, reliability coefficients were evaluated. Both alpha and omega coefficients supported the reliability of the PHQ and MDRS, with comparable values reported for each scale across the three time points (see Table 1 for MDRS subscales; Table 2 for total scores). Robust (*p*’s < 0.001) intercorrelations were observed between the PHQ and MDRS total scores ranging from moderate to strong associations.

### 3.4. Latent Growth Modelling

To examine the subsequent aims, we first conducted a MDLGM model to establish model fit and added current treatment at baseline as a time-invariant predictor. Linearity was confirmed, as we observed a decrement to the chi-square value for the competing quadratic (e.g., curved) model. As significant associations were expected between modelled slope and intercept values for the PHQ and MDRS, these terms were allowed to correlate in the model. Initial model fit indices indicated that there was need for model improvement (CFI = 0.954, TLI = 0.902, RMSEA = 0.182, SRMR = 0.0259). Significant covariance estimates were observed between the PHQ and MDRS intercepts and slopes, indicating that these variables tended to vary in similar ways across the time points. Modification indices were inspected, indicating that a substantial parameter change would occur by freeing (e.g., correlating) the error terms for the PHQ and MDRS at the 3-month time point. Atheoretical post-hoc model re-specification should be avoided, as it risks incorrect model specification [44]. However, given that previous research has highlighted the significant positive longitudinal association between the PHQ and MDRS [27], there was a rationale for permitting the error estimates for these variables to correlate, especially given the association between these constructs within the same (e.g., 3-month) time point [45].

Addressing Aim 2, the re-specified model yielded excellent fit statistics (CFI = 0.992, TLI = 0.980, RMSEA = 0.070, SRMR = 0.0281), which became the basis of interpretation and further analysis. The slope values for both the PHQ (−1.376, *p* < 0.001) and MDRS (−3.917, *p* < 0.001) were negative, showing that scores for both the PHQ and MDRS, on average, decreased between baseline and 6 months (e.g., symptoms marginally improved, with scores becoming less severe by one point on the PHQ and almost four points on the MDRS).

When the within-domain covariance was examined (e.g., covariance between the intercept and slope related to the same construct), the estimated covariance between the intercept and slope factors for PHQ was not statistically significant (*p* = 0.193). This indicated no difference in the PHQ rate of change between baseline and 6 months relative to baseline PHQ scores. In contrast, the estimated covariance between the intercept and slope factors for MDRS was statistically significant (*p* < 0.001). The negative estimate value (−66.720) suggests that men whose MDRS scores were high at baseline demonstrated a lower rate of change in these scores over the 6-month period than was the case for men whose MDRS scores were lower at Time 1 (even though, on average, MDRS scores went down over time, men with higher baseline MDRS scores improved less quickly than men with lower MDRS scores). Turning to the first between-domain covariance (MDRS slope/PHQ slope), a very strong relationship between the standardised coefficients (*r* = 0.823; *p* < 0.001) indicated a longitudinal association between the MDRS and PHQ (as men’s MDRS scores between baseline and 6 months underwent a strong decrease, so too did their PHQ scores). Similarly, the covariance for the PHQ and MDRS intercepts was also significant (r = 0.667; *p* < 0.001), indicating that men reporting higher MDRS scores also tended to have higher PHQ scores. These findings revealed robust inter-individual differences in both the initial scores of PHQ and MDRS at baseline and their change over 6 months. Such evidence of inter-individual differences provides powerful support for further investigation of variability related to the growth trajectories [39], in particular, the incorporation of predictors into the model to explain variability.

### 3.5. Effect of Current Treatment

Provided with evidence of inter-individual differences, we then asked whether, and to what extent, current treatment might explain this heterogeneity. In particular, we asked if PHQ and MDRS scores differed for those who were either currently accessing or not accessing treatment (Aim 3). Additionally, we asked if trajectories of change for the PHQ and MDRS varied as a function of experiencing mental health problems either with or without current treatment (Aim 4). The subsequent model, including the predictor of baseline current treatment, reported good model fit χ^2^(8) = 20.23, *p* = 0.010, CFI = 0.964, TLI = 0.986, RMSEA = 0.081, SRMR = 0.033 (see Figure 1).

Table 3 shows that current treatment was not a statistically significant predictor of PHQ scores at baseline (−1.202, *p* = 0.145), but current treatment did predict PHQ rate of change (0.897, *p* = 0.033). Given a coding of 0 for ‘no current treatment’ and 1 for ‘current treatment,’ these findings suggest that the rate of change was faster (by 0.897 PHQ points over 6 months) for those reporting baseline current treatment than for those reporting no current treatment. Results for the MDRS indicated that current treatment was a statistically significant predictor of both initial MDRS severity (−7.476, *p* = 0.006) and rate of MDRS change (2.749, *p* = 0.018). These findings suggest that MDRS scores were lower (e.g., better) for men reporting current treatment, and men reporting current treatment reported a faster rate of improvement on MDRS scores by 2.749 points over the 6-month period compared to men not accessing treatment. When current treatment at 3 and 6 months were introduced as time variant predictors, the multidomain model reported very poor model fit according to all indices (CFI = 0.886, TLI = 0.773, RMSEA = 0.172, SRMR = 0.129). This indicated that the present dataset was unable to test the longitudinal impact of treatment, a likely function of the comparatively small sample size for complex SEM models.

In terms of the effect magnitude (*R*^2^ values), while still proportionally low, current treatment accounted for over three-times the variance in MDRS intercept values (3.4%) than it did for PHQ intercept values (1.1%). There was no differentiation for current treatment on the rate of change for PHQ or MDRS (3.2%, respectively). For baseline PHQ scores, 76.0% of the variance was accounted for by predictors (e.g., by the intercept, slope, and current treatment). In contrast, 92.0% of the variance in baseline MDRS scores was explained. This indicates that the MDRS trajectory model was better able to account for baseline MDRS scores than the PHQ trajectory model was for predicting baseline PHQ scores. At 3 and 6 months, the proportion of variance accounted for was largely equivalent between the PHQ (3 months: 64.1%; 6 months: 86.2%) and MDRS (3 months: 65.5%; 6 months: 88.3%). This shows that the PHQ and MDRS appeared to have similar measurement utility over time, while also suggesting that due to the higher *R*^2^ value at baseline, the MDRS may better identify men’s symptom domains relative to current treatment compared to the PHQ.

## 4. Discussion

Longstanding commentaries and emergent empirical work have highlighted the possibility that men’s depression may be missed clinically as a by-product of residing outside generic screens (e.g., the PHQ), which may be insensitive to men’s socialisation processes and internalised traditional gender norms [46,47]. By engaging a comparison of the MDRS and PHQ longitudinally in a sample of men who were in and out of treatment, the current study made available critically important clinical considerations. The PHQ is a widely used and validated measure of prototypic depression symptoms [32] and is therefore an important point of comparison for the MDRS. In this study (and consistent with prior work [27]), both scales reported satisfactory internal consistency across the three waves of data for the alpha, with more rigorous omega reliability coefficients. The finding that reliability coefficients between the two scales were equivalent indicates that the MDRS and PHQ consistently measure their target constructs. That said, in comparison to the PHQ, the MDRS may have greater sensitivity to detecting change for men currently in treatment (and who were reporting a mental health (MH) problem) compared to those not in treatment (and who were reporting a MH problem). Supporting the utility and responsiveness to treatment, for both PHQ and MDRS, improvement was faster for men reporting a mental health problem who were currently accessing treatment. Of note, the baseline between-group analysis indicated no difference for PHQ scores but a significant difference for MDRS scores relative to current treatment, which suggests that the MDRS may be better able to differentiate men’s treatment response to mental health intervention than the PHQ. Nonetheless, this is a finding that needs to be replicated in future work.

Broadly speaking, mean baseline MDRS subscale and item scores were higher in the present sample compared to prior MDRS research undertaken with the general population [28]. This is perhaps unsurprising given that the participants in the present sample were seeking information on men’s depression via the *HeadsUpGuys* website [33], and were therefore more likely to be symptomatic compared to men in the general population. Of the six MDRS subscales, the emotion suppression and risk-taking domains were significantly higher at baseline for men not accessing treatment than treatment-engaged men. At the individual MDRS item level, these effects appeared largely driven by three items assessing stoicism (e.g., working things out independently), and recklessness (e.g., stopping caring about consequences of actions, taking unnecessary risks), though nonsignificant univariate trends (*p* < 0.10) for higher scores in men not engaged in treatment were also observed for items assessing bottling up negative feelings, overreaction with aggressive behaviour and requiring drugs to cope and somatic symptoms (e.g., heartburn, aches). Though speculative, it may be the case that these domains serve as treatment barriers in their own right by men trying to avoid or suppress uncomfortable emotions deliberately (a cognitively demanding state that confers health risks [48,49]) or alternatively through distraction routines that may co-occur with risk taking behaviours (possibly as means of enacting a sense of control [9]). The MDRS total score subsumes these domains, which are strongly correlated with the PHQ, yet also distinct from the internalising prototypic depression symptoms that are assessed by the PHQ (e.g., anhedonia, sadness, guilt), which is a strength of the scale in providing a broader perspective on men’s depression or distress.

When the dual growth curve models were evaluated without current treatment as a predictor, the within-domain covariance indicated that there was no difference on the PHQ rate of change between baseline and 6 months relative to the baseline PHQ score. In contrast, the MDRS slope was significant (*p* < 0.001) with a negative estimate value, indicating that men with higher baseline MDRS scores improved slower than men with lower MDRS scores. This shows that MDRS severity at baseline (but not PHQ severity) reduces the rate of change that can be expected, indicating that when MDRS domains are more severe, change is harder to achieve. This finding suggests a potential differential responsiveness to treatment assessed by the PHQ and MDRS, especially for those men at the severe end of the scales. The strong between-domain covariance (*r* = 0.823, *p* < 0.001) indicated that the trajectory of MDRS score changes co-occurred with PHQ changes, and the moderate-strong intercept correlation indicated that men reporting higher MDRS scores also reported higher PHQ scores. These findings show that the MDRS and PHQ domains ‘travel together in time,’ supportive of the putative function of domains assessed by the MDRS that place men at risk of major depression.

The observed inter-individual variation justified the inclusion of a predictor to explore potential group differences. Given that the entire sample self-indicated the presence of a mental health problem, engagement in current treatment was considered an important variable to explore. Goodness-of-fit indices indicated that including current treatment as a predictor resulted in excellent model fit (although if the more stringent criteria of RMSEA < 0.08 is applied [39], the RMSEA value could be considered marginal), highlighting the importance of this variable in accounting for the observed inter-individual variation. Results indicate that men currently in treatment reported significantly lower MDRS scores at baseline, but they did not report significantly lower PHQ scores at baseline. This is of note, as the initial dual model indicated that unlike PHQ scores, men with higher MDRS scores experienced less improvement over time. Whereas the present data did not allow us to identify how long men were in treatment (which may impact PHQ and MDRS scores), the models included three waves of data from all participants, and the MDLGM approach accounted for inter-individual difference. Regardless of the amount of time men had received current treatment, those currently in treatment tended to have lower MDRS scores, but not lower PHQ scores than men reporting a mental health problem and not in treatment. Therefore, while MDRS severity resulted in less change over time, MDRS domains appeared amenable to intervention, and this change can be assessed by the scale. Further, when examining the slope statistics (e.g., improvement over time), both the PHQ and MDRS improved more rapidly for men currently in treatment compared to those not in treatment (again, this shows that both scales are able to detect change associated with current treatment). These findings support application of the MDRS in clinical settings.

A range of study limitations and future directions should be considered. Whereas the present sample was strengthened by the use of three waves of data, it was limited by size, as samples of 200 are considered the minimum for valid growth curve analyses. Nonetheless, we observed a robust correlation between the PHQ and MDRS slopes, and Lee and Whittaker [37] suggest that researchers can be confident in statistically significant group differences when effect sizes are at least moderate (e.g., 0.50) in sample sizes as small as 200. That said, if the sample exceeded 400, then we could expect to have observed sufficient power to estimate time variant effects and potentially a more favourable RMSEA value [41]. It is also important to bear in mind that results may be biased as a function of the comparatively small (8.3%) proportion of study respondents from the larger baseline sample (*n* = 3769) who identified a mental health problem and provided data at waves 2 and 3. Our sampling method also introduced the risk of bias, given participants visiting the *HeadsUpGuys* site were help-seeking or proactively seeking information on men and depression. This limits the generalisability of findings to non-help-seeking populations. Nonetheless, severity of prototypic depression symptoms was equivalent at baseline between those accessing and not accessing mental health support. Hence, group effects were not due to differences in depressive severity. Furthermore, while analyses explored whether men were engaged in current treatment at baseline, model fit indices indicated that inclusion of current treatment as a time variant covariate at 3 and 6 months was yielded a poor fit to the data. We were unable to determine whether this was a function of sample size (which we believe as the likely explanation given the good model fit achieved when baseline current treatment was a covariate), or was instead suggestive that the longitudinal modelling of current treatment inadequately explains differing growth trajectories for the PHQ or MDRS. Again, future longitudinal studies drawing on larger samples are therefore needed. In the present study, participants self-identified as having a mental health problem. This was not validated by clinical interview or diagnosis. Further, the actual mental health problem(s) that men were referring to was not captured, and data were not captured on the severity or duration of this problem or treatment modality accessed. All participants were recruited via the *HeadsUpGuys* website which provides men’s depression psychoeducation. Results should be verified in a broader population of men who are not necessarily actively seeking mental health information. It is recommended that further psychometric evaluation of the MDRS be undertaken, including invariance testing and the further establishment of a hierarchal factor structure model where the six MDRS domains load on a latent factor for male depression risk, e.g., [26], or testing a unidimensional model of the MDRS regarding use of the scale total score, e.g., [50]. At present, the MDRS uses an 8-point response scale, and as a 22-item tool, it may be too lengthy for use in primary care settings. Efforts are currently underway to validate an MDRS short form using a condensed response format, in addition to cross-nation validation and translation [51]. These developments may increase the likelihood of the MDRS being used adjunctively with brief prototypic depression measures such as the PHQ.

The construct of intersectionality and its application to the field of men’s mental health is growing [52]. Intersectionality focuses on the connecting and overlapping aspects of personhood and identity [53] and is increasingly used in the field of gender and health internationally [54]. Better understanding the ways in which prototypic and male-specific symptoms of depression intersect with domains of sexuality, social class, race/ethnicity and income, and corresponding links to maladaptive behaviours and suicide risk is an important future endeavour for the field [55]. Finally, given that the present data was collected from a help-seeking population, it was not suited to evaluating whether the MDRS is able to detect a subgroup of men that may be missed on the PHQ but who go on to develop a major depressive illness. An answer to this question is needed to evaluate the true value and utility of the MDRS as (i) a measure of a potential prodromal depression state for men or (ii) conclusively determining whether the putative male depression subtype exists.

## 5. Conclusions

The present study provides important psychometric information on the MDRS. Findings showed that in a symptomatic help-seeking population of men, the MDRS exhibited comparable reliability to the widely used PHQ. Whereas robust correlations were observed between the two scales, they assessed distinct domains relevant to men’s mental health. Relative to the PHQ, the MDRS performed well in terms of longitudinal sensitivity to change. Results support the ongoing use of the MDRS, including validation in diverse cultures and samples. Male-specific measures such as the MDRS may improve the detection of depression in men, and adjunctive use (alongside established scales such as the PHQ) may contribute to improved public health outcomes.

## Figures and Tables

**Figure 1 ijerph-17-07322-f001:**
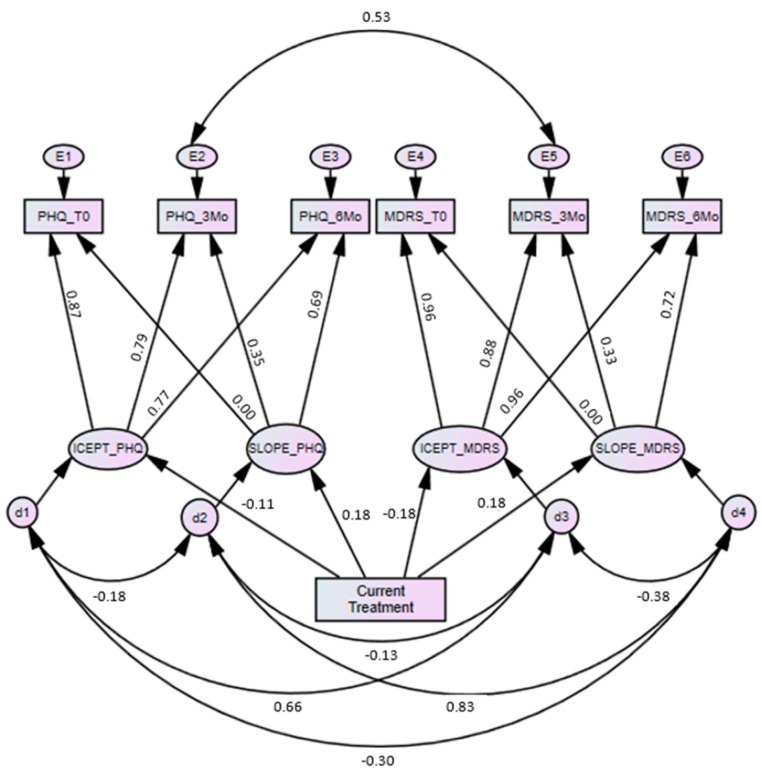
Multidomain latent growth model with current treatment as a predictor (standardized coefficients).

**Table 1 ijerph-17-07322-t001:** Male Depression Risk Scale (MDRS) 22-item and subscale group differences (baseline).

		Current Treatment *n* = 133	No Current Treatment *n* = 101	*F*, *p*-Value, Effect Size (η^2^)
MDRS Item	MDRS Subscale	*M* (*SD*)	*M* (*SD*)	
1. I bottled up my negative feelings	Emot Supp	4.62 (1.99)	5.14 (1.96)	3.77, 0.053, (0.016)
2. I covered up my difficulties	Emot Supp	4.83 (1.99)	5.25 (1.81)	2.69, 0.102, (0.012)
3. I drank more alcohol than usual	Alcohol	1.04 (1.71)	1.32 (1.96)	1.15, 0.284, (0.005)
4. I drove dangerously or aggressively	Risk	0.71 (1.42)	1.04 (1.87)	2.29, 0.132, (0.010)
5. I had more heartburn than usual	Somatic	1.04 (1.56)	1.50 (2.00)	3.48, 0.063, (0.015)
6. I had regular headaches	Somatic	1.94 (2.12)	1.70 (1.99)	1.03, 0.312, (0.004)
7. I had stomach pains	Somatic	1.76 (2.14)	1.80 (2.13)	0.02, 0.893, (0.000)
8. I had to work things out by myself	Emot Supp	4.77 (1.88)	5.71 (1.72)	15.02, <**0.001**, (0.061)
9. I had unexplained aches and pains	Somatic	2.43 (2.40)	2.92 (2.56)	3.72, 0.055, (0.016)
10. I needed alcohol to help me unwind	Alcohol	1.11 (2.09)	1.35 (2.18)	0.62, 0.431, (0.003)
11. I needed to have easy access to alcohol	Alcohol	0.81 (1.91)	0.83 (1.87)	0.00, 971, (0.000)
12. I overreacted to situations with aggressive behaviour	Anger	1.71 (1.84)	2.19 (1.93)	3.83, 0.052, (0.016)
13. I sought out drugs	Drug Use	1.05 (2.08)	1.24 (2.29)	0.54, 0.461, (0.002)
14. I stopped caring about the consequences of my actions	Risk	1.44 (1.82)	2.18 (2.01)	7.02, **0.009**, (0.029)
15. I stopped feeling so bad while drinking	Alcohol	0.93 (1.86)	1.24 (1.97)	1.08, 0.300, (0.005)
16. I took unnecessary risks	Risk	0.98 (1.56)	1.50 (1.80)	4.32, **0.039**, (0.018)
17. I tried to ignore feeling down	Emot Supp	3.89 (2.29)	4.49 (2.46)	2.64, 0.106, (0.011)
18. I used drugs to cope	Drug Use	0.95 (2.07)	1.46 (2.54)	3.03, 0.083, (0.013)
19. I verbally lashed out at others without being provoked	Anger	1.26 (1.68)	1.45 (1.58)	0.83, 0.363, (0.004)
20. I was verbally aggressive to others	Anger	1.28 (1.62)	1.61 (1.69)	2.41, 0.012, (0.010)
21. It was difficult to manage my anger	Anger	1.93 (2.06)	2.48 (2.29)	3.10, 0.808, (0.013)
22. Using drugs provided temporary relief	Drug Use	0.92 (1.97)	1.20 (2.20)	0.99, 0.320, (0.004)
**Subscale/Total Score**	**α (ω)**			
MDRS- Emotion Suppression	0.732 (0.729)	18.12 (6.24)	20.58 (5.60)	8.91, **0.003**, (0.037)
MDRS—Drug Use	0.953 (0.956)	2.92 (5.91)	3.90 (6.67)	1.50, 0.223, (0.006)
MDRS—Alcohol Use	0.932 (0.936)	3.89 (7.00)	4.73 (7.34)	0.65, 0.420, (0.003)
MDRS—Anger & Aggression	0.856 (0.856)	6.19 (6.19)	7.72 (6.30)	3.51, 0.062, (0.015)
MDRS—Somatic Symptoms	0.776 (0.781)	7.66 (6.93)	7.42 (6.19)	0.26, 0.871, (0.032)
MDRS—Risk Taking	0.652 (0.695)	3.13 (3.99)	4.72 (4.13)	7.69, **0.006**, (0.032)
MDRS—22 total score	See Table 2	41.90 (21.50)	49.08 (19.93)	6.35, **0.012**, (0.027)
PHQ-9 total score	See Table 2	16.06 (6.52)	17.12 (6.07)	1.60, 0.206, (0.007)

Bold font denotes significant at *p* < 0.05; Emot Supp = Emotion Suppression; MDRS items are reprinted with permission from Taylor & Francis Ltd. From the publication Rice et al., *Journal of Mental Health* 2019, 28, (2), 132–140, www.tandfonline.com.

**Table 2 ijerph-17-07322-t002:** Internal consistency and Pearson coefficients across the three waves.

	PHQ			MDRS		
**Reliability**	**Baseline**	**3-mo**	**6-mo**	**Baseline**	**3-mo**	**6-mo**
Cronbach alpha	0.864	0.899	0.906	0.842	0.879	0.871
McDonald omega	0.861	0.897	0.905	0.806	0.861	0.852
**Pearson correlation**	**1.**	**2.**	**3.**	**4.**	**5.**	**6.**
1. PHQ baseline	-					
2. PHQ 3-Mo	0.705 ***	-				
3. PHQ 6-Mo	0.643 ***	0.713 ***	-			
4. MDRS baseline	0.557 ***	0.318 ***	0.360 ***	-		
5. MDRS 3-Mo	0.484 ***	0.602 ***	0.534 ***	0.599 ***	-	
6. MDRS 6-Mo	0.389 ***	0.434 ***	0.632 ***	0.549 ***	0.699 ***	-

*** denotes *p* < 0.001.

**Table 3 ijerph-17-07322-t003:** Model estimates.

Regression Weights	Estimate	SE	C.R.	*p*
PHQ intercept <- Current Tx	−1.202	0.825	−1.456	0.145
PHQ slope <- Current Tx	0.897	0.421	2.131	0.033
MDRS intercept <- Current Tx	−7.476	2.719	−2.749	0.006
MDRS slope <- Current Tx	2.749	1.164	2.361	0.018
**Standardised estimates**				
PHQ intercept <- Current Tx	−0.107			
PHQ slope <- Current Tx	0.179			
MDRS intercept <- Current Tx	−0.184			
MDRS slope <- Current Tx	0.179			
PHQ baseline <- PHQ intercept	0.872			
PHQ baseline <- PHQ slope	0.000			
PHQ 3-Mo <- PHQ intercept	0.792			
PHQ 3-Mo <- PHQ slope	0.353			
PHQ 6-Mo <- PHQ intercept	0.774			
PHQ 6-Mo <- PHQ slope	0.689			
MDRS baseline <- MDRS intercept	0.959			
MDRS baseline <- MDRS slope	0.000			
MDRS 3-Mo <- MDRS intercept	0.884			
MDRS 3-Mo <- MDRS slope	0.334			
MDRS 6-Mo <- MDRS intercept	0.956			
MDRS 6-Mo <- MDRS slope	0.723			

Tx = treatment SE = standard error, CR = critical ratio.

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
