# Peer review of "Male-Type and Prototypal Depression Trajectories for Men Experiencing Mental Health Problems"

_ijerph, 2020, doi:10.3390/ijerph17197322_

Round 1

Reviewer 1 Report

This study aimed at a significant topic. However, there are some methodological issues requiring major reviews: 

  1. Sampling methods are unclear, please provide a detailed description of sample selection - this is crucial to apply findings to other groups 
  2. "recruited online via the HeadsUpGuys" Recruitment methods are unclear and can evoke bias
  3. Due to the sampling methods (on-line questionnaire) the age of the participants is relatively low. This may have implications for the results. 
  4. Due to the methodological limitations, findings from this study should be clearly defined.

Author Response

We are grateful to Reviewer 1 for their constructive comments. Please refer to the uploaded document attached for our responses raised to reviewer comments, and changes made to the paper.

Reviewer 2 Report

I consider this manuscript to be ready for publication. My only observations are that 2 tables are labelled as "Table 2". Thus, "Table 2. Internal consistency and Pearson coefficients across the three waves" should remain being Table 2, while the table labeled as "Table 2. Model Estimates" should be re-labeled as Table 3. Also, in order to preserve consistency across the manuscript, Estimate Scores should be shown on the first data point reported on the paragraph shown on Page 7. I state this because Estimate Scores are shown later in the same sentence and throughout the rest of the paragraph for all other shown data points. Here I show the specific sentence I feel needs to have the suggested modification implemented: "Table 2 shows that current treatment was not a statistically significant predictor of PHQ scores at 238 baseline (p=.145), but current treatment did predict PHQ rate of change (0.897, p=.033)." Once these 2 suggestions gets implemented, the manuscript would be ready for publication.

Author Response

We are grateful to Reviewer 2 for their constructive comments. Please refer to the uploaded document for our responses raised to reviewer comments, and changes made to the paper.

Round 2

Reviewer 1 Report

The authors provided comprehensive explanations. Revisions were prepared strictly according to the suggestions. The current version of the manuscript is balanced and the methodology is precisely described.